# Alcohol outlet density and adolescent drinking behaviors in Thailand, 2007–2017: A spatiotemporal mixed model analysis

**Polathep Vichitkunakorn**[1], **Sawitri Assanangkornchai**[2], **Kanittha Thaikla**[3], **Suhaimee Buya**[4,5], **Supeecha Rungruang**[1], **Mfahmee Talib**[6], **Warangkhana Duangpaen**[7], **Warintorn Bunyanukul**[8], **Monsicha Sittisombut**[8]*

1 Department of Family and Preventive Medicine, Faculty of Medicine, Prince of Songkla University, Hat Yai, Songkhla, Thailand, 2 Department of Epidemiology, Faculty of Medicine, Prince of Songkla University, Hat Yai, Songkhla, Thailand, 3 Research Institute for Health Sciences, Chiang Mai University, Chiang Mai, Thailand, 4 School of Information, Computer and Communication Technology, Sirindhorn International Institute of Technology, Thammasat University, Bangkok, Thailand, 5 School of Knowledge Science, Japan Advanced Institute of Science and Technology, Ishikawa, Japan, 6 Faculty of Nursing, Prince of Songkla University, Pattani Campus, Muang, Pattani, Thailand, 7 Department of Mathematics and Statistics, Faculty of Science, Prince of Songkla, University, Songkhla, Thailand, 8 School of Medicine and Health Sciences, Faculty of Medicine, Prince of Songkla University, Hat Yai, Songkhla, Thailand

* 6210310175@psu.ac.th

## Abstract

This study aimed to explore the relationship between alcohol outlet density and the proportion of alcohol consumption among Thai adolescents. We utilized the alcohol consumption data from the 2007, 2011, and 2017 Tobacco and Alcohol Consumption Survey in Thailand. We analyzed the relationships between alcohol sales license figures and consumption behavior using a spatiotemporal mixed model. Our data had two levels. The upper (provincial) level featured alcohol sales license density (spatial effect), the years of survey (temporal effect), and the social deprivation index. The lower (individual) level included the demographic data of the adolescents. A total of 9,566 Thai adolescents participated in this study, based on surveys conducted in 2007 (n = 4,731), 2011 (n = 3,466), and 2017 (n = 1,369). The density of alcohol sales licenses increased the odds for the one-year current drinker category (odds ratio [OR] = 1.08, 95% confidence interval [CI], 1.04–1.45), especially in male adolescents (OR = 1.09, 95% CI, 1.04–1.14). Furthermore, it increased the odds for the heavy episodic drinker category for female adolescents (OR = 1.23, 95% CI, 1.05–1.44). Increased alcohol sales licenses are associated with higher alcohol consumption among Thai adolescents. This highlights the need for government organizations to develop and apply strategies to reduce the number of licenses for the sale of alcohol.

## Introduction

Alcohol is a major threat to the individual and society as a whole. The effects on individual health range from cardiovascular diseases and infectious diseases to injuries [1]. It can lead to

**Data Availability Statement:** The dataset was obtained with permission from Thailand's National

Statistical Office (https://www.nso.go.th/) and Excise Department (https://www.excise.go.th/). Due to ethical limitations governing the dissemination of a de-identified dataset, it is not permissible to share the data publicly. Interested researchers are welcome to contact the corresponding author, Monsicha Sittisombut (6210310175@email.psu.ac.th) to facilitate discussions regarding their requests for confidential data access. Furthermore, researchers can contact the National Statistical Office and the Excise Department directly via services@nso.go.th and contact@excise.go.th, respectively, to obtain permission to access the dataset.

**Funding:** This research was funded by the Health Systems Research Institute Thailand (http://www.hsri.or.th). The grant number was 64-003. The funder did not play any role in the study design, data collection and analysis, decision to publish, nor preparation of the manuscript.

**Competing interests:** The authors have declared that no competing interests exist.

suicide and interpersonal violence [2]. Specifically, alcohol consumption in adolescents can lead to cognitive function deficit [3], an increased risk of psychological distress [4], development of severe liver disease [5], and alcohol dependence in adulthood [6]. To exacerbate the situation, there has been an increase in various marketing efforts from the alcohol industry, including mass media commercials, in-store displays, merchandise, and online marketing [7, 8].

The World Health Organization has introduced SAFER strategies as alcohol control interventions to reduce harmful alcohol use. The strategies consist of the following: *(1) Strengthen restriction on alcohol availability; (2) Advance and enforce drunk driving countermeasures; (3) Facilitate access to screening, brief interventions, and treatment; (4) Enforce bans or comprehensive restrictions on alcohol advertising, sponsorship, and promotion; and (5) Raise prices on alcohol through excise taxes and pricing policies* [9].

Currently, Thailand has three major national laws that are in accordance with the SAFER strategies: the Alcoholic Beverage Control Act B.E. 2551 (2008), Excise Act B.E. 2560 (2017), and Road Traffic Act B.E. 2522 (1979) [10]. The Alcoholic Beverage Control Act B.E. 2551 (2008) is Thailand's first legislative effort aimed specifically at controlling the consumption and distribution of alcoholic beverages [11]. The Excise Act B.E. 2560 (2017) regulates the taxation and licensing of production, and the sales and import of products and services, including alcoholic products [12]. In the context of alcohol regulation, the Road Traffic Act B.E.2522 (1979) sets limits on the alcohol blood content (BAC) for drivers [10].

The alcohol industry targets adolescents [7]. Thailand's National Statistical Office reported in the 2021 survey that 11.6% of participants aged 15–19 had consumed an alcoholic beverage at least once in their lifetime [13] despite the fact that Section 29 of the Alcoholic Beverage Control Act B.E. 2551 (2008) [11] prohibits the sale of alcoholic beverages to a person under the age of 20. Moreover, a study has shown that Thai adolescents are inclined to start drinking earlier than in the past [14].

In addition to causing health problems, alcohol consumption is one of the risk factors for road accidents [15, 16]. Road accidents are more common among adolescents, and are associated with alcohol use [17–23]. According to the World Health Organization, Thailand has the highest number of road accidents in the Association of Southeast Asian Nations and ranks in the top 10 worldwide, primarily owing to speed and binge drinking [24].

Restriction of physical availability, which is a part of SAFER strategies, has been proven effective [2]. Physical availability refers to the availability of alcohol within a person's immediate surroundings, influenced by the likelihood that a person can encounter alcohol outlets [25]. The higher the alcohol outlet density, the easier it is for adolescents to access alcohol [22]. The density of alcohol sale outlets can be monitored using the "density of alcohol sales licenses" per 1,000 people. In the Thai context, the density of alcohol licenses can be determined from the number of sales licenses registered with the Excise Department. According to the Excise Act B.E. 2560 (2017), any person who intends to sell liquor needs permission, in the form of an alcohol sales license, from the excise authority. A study in New Zealand [26] examined the relationship between alcohol consumption and licenses granted within a 3 km radius of six university campuses and found that alcohol license density was positively correlated with drinkers and alcohol-related problems as well as alcohol consumption among adolescents.

Although a growing body of studies has documented the relationship between alcohol license density and alcohol consumption behaviors in adolescents, there remains a marked gap in the literature regarding the effects in the context of developing countries. The existing studies were predominantly single-center studies conducted mostly in developed countries [27, 28]. A critical review, focusing primarily on literature from developed countries, highlighted

the need for diverse research settings as different jurisdictions have varying norms and legislation [29]. Furthermore, much of the existing research in developing countries has been limited due to the methodological constraints. For instance, they are mostly based on single-year data [30–32]. Moreover, most studies have used longitudinal or panel data to track temporal changes [33], with outcomes including intimate partner violence [34–37], drunk driving [38], motorcycle accidents [39], and hospitalization rates [33], rather than focusing directly on alcohol consumption as the outcome.

To address this gap in knowledge, we aimed to examine the relationship between alcohol license density and the proportion of alcohol consumption among adolescents at the provincial level from 2007 to 2017 in Thailand. Our hypothesis was as follows: the higher the alcohol density, the higher the proportion of adolescents' alcohol use. Our findings will help bridge the gap between existing research in developed and developing countries as well as inform policymakers of measures for controlling liquor licenses in Thailand.

## Materials and methods

### Study design

This was a cross-sectional study involving an analysis of secondary data: nationally representative surveys and pre-existing sales license data. For the alcohol consumption data, we utilized data from the 2007, 2011, and 2017 waves of the Tobacco and Alcohol Consumption Surveys. They are nation-wide alcohol consumption surveys that have been conducted every four years by Thailand's National Statistical Office since 2001.

For the alcohol outlet data, we employed the aggregate data on the number of alcohol sales licenses, involving the total number of licenses approved by Thailand's Excise Department, Ministry of Finance, for each province in the years 2007, 2011 and 2017. The dataset also included the annual statistical information report on the population and households from the official statistics registration systems of the Department of Provincial Administration, Ministry of the Interior, Thailand.

The authors have applied for an amendment regarding protocols for analyses of anonymized secondary data. All obtained data were anonymous. Ethical approval was obtained from the Human Research Ethics Committee of the Faculty of Medicine, Prince of Songkla University (REC. 62-054-18-1).

### Participants

Our sample population, aged 15–19 years, was extracted from the original surveys, which recruited participants aged 15 years and older. Of the three waves, the survey in 2007 involved data collection from 4,731 adolescents. In the subsequent wave in 2011, 3,466 adolescents were recruited. The most recent wave in 2017 included 1,369 participants. Cumulatively, these surveys represent a total of 9,566 Thai adolescents. We acknowledge that our study includes participants below the legal drinking age.

### Data collection

We obtained the secondary data concerning alcohol consumption and alcohol sales licenses directly from the Thai National Statistical Office and the Excise Department, respectively. Originally, all alcohol consumption survey waves employed a two-stage stratified sampling approach. The provinces were chosen as strata. The units for the first and second stages were villages and households, respectively. The villages were chosen in proportion to the population of each province. The households were systematically selected. However, notably, despite these

methodological differences, the survey population and data collection procedures remained consistent across all three waves. In each instance, data were collected through face-to-face interviews, encompassing individuals aged 15 years and older from all geographical regions, provinces, and districts within Thailand. All data were accessed online on January 11, 2021.

**Dependent variable: Proportion of alcohol consumption.** The outcome variable was proportion of alcohol consumption, including the proportion of current drinkers, regular drinkers, and heavy episodic drinkers (HEDs). To categorize the drinkers, we collected drinking frequency and the quantity of alcohol consumed as variables. The proportion was calculated as follows:

- We calculated the proportion of *current drinkers* as the number of participants who drank at least one standard drink of alcohol during the 12 months preceding the interview and divided by the number of all participants.

- We calculated the proportion of *regular drinkers* using the number of current drinkers who drank alcohol at least once per week or more, divided by the number of all current drinkers. Regular drinking is related to the lifetime risk of hospitalization for alcohol-related problems [40, 41].

- We calculated the proportion of *HEDs* by dividing the number of current drinkers—those who drank at least four to five standard alcoholic drinks on at least one occasion—by the number of all current drinkers. This is one of the most important indicators of acute alcohol-related harm (e.g., injuries, accidents, and acute social consequences) [42] and chronic diseases (e.g., tuberculosis, epilepsy, ischemic heart disease, and cirrhosis) [43]. These can be provided by alcohol intoxication and drinking intensity [44].

**Independent variables: Density of alcohol outlets or alcohol sales licenses.** In this study, we measured alcohol outlet density by alcohol sales license density and by population size, following the recommendations of the United States Department of Health and Human Services [45]. This index reflects the density of alcohol sales licenses in each province. We used the following formula:

$$Density\ of\ alcohol\ sales\ licenses\ compared\ with\ the\ population = \frac{Number\ of\ alcohol\ sales\ licences\ in\ each\ province}{Midyear\ population\ in\ each\ province} \quad (1)$$

**Potential confounding variable: Social deprivation index and other demographics.** The potential confounding variables include the social deprivation index, survey year, region of residence, household area, number of household members, marital status, educational level, monthly household income, and smoking status. The social deprivation index (SDI) is a socioeconomic indicator of social disparity, derived by integrating 18 economic and social variables from eight domains (location, demography, education, disability, employment, housing, crowdedness, and residential mobility). The relationships among the variables were explained using principal component analysis. We applied the SDI for each province in Thailand [46]. We then categorized all provinces into five equal SDI groups (quintiles). In the fifth quintile, the group with the highest values was the poorest. The group with the lowest value (first quintile) was the wealthiest. All confounding variables were collected and analyzed for each participant.

## Statistical analysis

We applied a spatiotemporal mixed model using the following formula. The spatiotemporal mixed model forms a two-level nested structure by imposing alcohol consumption—including

current, regular, and HEDs—as dependent variables. The independent variables contained two levels of information. At the upper (provincial) level, the major independent variables were the following: density of alcohol sales licenses in each province (spatial effect), survey year (temporal effect), and SDI. This model allowed us to analyze how the temporal dynamics —reflected through the survey years—impact alcohol consumption patterns across different provinces, alongside the spatial distribution of alcohol sales licenses. The confounding variables were collected at the lower (individual) level. They include survey year, region of residence, SDI, household area, number of household members, marital status, educational level, monthly household income, and smoking status.

$$Logit\ (odds) = B_{00} + (B_{10} + u_{1j}) * X_{ig} + u_{oj} \tag{2}$$

where: $B_{00}$ is a constant

$B_{10}$ is a fixed slope

$u_{1j}$ is the deviation of the cluster-specific slope from the fixed slope

$u_{0j}$ is the random intercept variance

$X_{ij}$ represents the observations of the independent variable sequence I to group j

We applied a heatmap to visualize the distribution of the dependent (density of alcohol sales licenses in each province) and independent variables (proportion of alcohol consumption). We used different colors on the heatmap to show the statistical values in the different provinces. The colors illustrate the density of drinkers and their behavior patterns; dark colors indicate a high proportion of adolescent alcohol consumption, and light colors indicate a low proportion.

We analyzed the data using R software and the "openxlsx," "epiDisplay," "epicalc," "data.table," "DescTools," "MASS," "lme4," "MCMCglmm," and "plyr" contributed packages. We created the map using "QGIS," an open source Geographic Information System (GIS) [47].

## Results

### Demographic data

The study surveyed Thai adolescents in 2007 (n = 4,731), 2011 (n = 3,466), and 2017 (n = 1,369), cumulatively comprising 9,566 participants (**Table 1**). The proportion of current drinkers increased from 24.4% in 2007 to 33.2% in 2017. The proportion varied significantly by region, from 38.3% in the north-east to 15.4% in the south. Current drinking rates were higher among the poor quintiles, small households, singles, and especially smokers.

For HEDs, the data show a significant fluctuation over the survey years ($p < 0.001$), with 79.3% of current drinkers experiencing heavy episodic drinking in 2007, 80.5% in 2011, and 45.2% in 2017. Heavy episodic drinking rates also varied by region. The proportion of HEDs significantly varied by household size ($p < 0.001$), marital status ($p = 0.097$), and smoking status ($p < 0.001$), with smokers having a significantly higher rate (77.0%).

For regular drinkers, there was a stable trend over the survey years, with no significant variation between regions, SDI quintiles, household area, educational level, or monthly household income ($p > 0.05$). Notably, smoking status significantly influenced regular drinking behavior, with smokers having a higher rate (45.7%, $p < 0.001$) compared to non-smokers.

### Trend of alcohol sales licenses in 2007–2018 in Thailand

**Fig 1** shows the number of alcohol sales licenses per 1,000 people from 2007 to 2018, an average of nine. In 2007 and 2008, the density of alcohol sales licenses was lower than in other years, while in 2015, it was higher than in other years. We concluded that the number of alcohol sales licenses showed steady density adjustments and no decline.

**Table 1. Percentage of alcohol drinking behavior with demographic and socioeconomic factors among Thai adolescents (n = 9,566).**

| Factors | Total | Alcohol drinking behavior (n = 9,566) | | |
|---|---|---|---|---|
| | | Current drinker, n (% of total) | Heavy episodic drinker, n (% of current drinkers) | Regular drinker, n (% of current drinkers) |
| Survey year | | | | |
| 2007 | 4731 | 1152 (24.4) | 914 (79.3) | 464 (40.3) |
| 2011 | 3466 | 1032 (29.8) | 831 (80.5) | 393 (38.1) |
| 2017 | 1369 | 454 (33.2) | 205 (45.2) | 164 (36.1) |
| | | $p < 0.001$* | $p < 0.001$* | $p = 0.266$ |
| Region | | | | |
| Bangkok | 466 | 120 (25.8) | 85 (70.8) | 52 (43.3) |
| Central | 3458 | 836 (24.2) | 592 (70.8) | 334 (40.0) |
| Northeastern | 2097 | 803 (38.3) | 617 (76.8) | 304 (37.9) |
| Northern | 1600 | 579 (36.2) | 442 (76.3) | 232 (40.1) |
| Southern | 1945 | 300 (15.4) | 214 (71.3) | 99 (33.0) |
| | | $p < 0.001$ | $p = 0.025$* | $p = 0.168$ |
| Social deprivation index | | | | |
| Q1 (richest) | 2460 | 625 (25.4) | 453 (72.5) | 252 (40.3) |
| Q2 | 1963 | 488 (24.9) | 370 (75.8) | 209 (42.8) |
| Q3 | 2069 | 561 (27.1) | 413 (73.6) | 211 (37.6) |
| Q4 | 1486 | 485 (32.6) | 358 (73.8) | 179 (36.9) |
| Q5 (poorest) | 1588 | 479 (30.2) | 356 (74.3) | 170 (35.5) |
| | | $p < 0.001$* | $p = 0.799$ | $p = 0.125$ |
| Household area | | | | |
| Urban | 4919 | 1346 (27.4) | 980 (72.8) | 534 (39.7) |
| Rural | 4647 | 1292 (27.8) | 970 (75.1) | 487 (37.7) |
| | | $p = 0.631$ | $p = 0.185$ | $p = 0.297$ |
| Number of household members | | | | |
| 1–2 | 1191 | 425 (35.7) | 237 (55.8) | 171 (40.2) |
| 3–4 | 4277 | 1235 (28.9) | 945 (76.5) | 463 (37.5) |
| 5–6 | 3052 | 756 (24.8) | 591 (78.2) | 293 (38.8) |
| >7 | 1046 | 222 (21.2) | 177 (79.7) | 94 (42.3) |
| | | $p < 0.001$* | $p < 0.001$* | $p = 0.489$ |
| Marital status | | | | |
| Single | 7107 | 2032 (28.6) | 1515 (74.6) | 802 (39.5) |
| Married | 2302 | 580 (25.2) | 420 (72.4) | 212 (36.6) |
| Divorced/separated | 157 | 26 (16.6) | 15 (57.7) | 7 (26.9) |
| | | $p < 0.001$* | $p = 0.097$ | $p = 0.207$ |
| Educational level | | | | |
| Primary school or below | 3300 | 872 (26.4) | 649 (74.4) | 359 (41.2) |
| High school | 5765 | 1623 (28.2) | 1200 (73.9) | 610 (37.6) |
| Bachelor's and above | 501 | 143 (28.5) | 101 (70.6) | 52 (36.4) |
| | | $p = 0.184$ | $p = 0.631$ | $p = 0.181$ |
| Monthly household income (THB/month) | | | | |
| ≤ 15000 | 9471 | 2602 (27.5) | 1926 (74.0) | 1006 (38.7) |
| 15001–30000 | 73 | 27 (37.0) | 16 (59.3) | 11 (40.7) |
| 30001–50000 | 13 | 6 (46.2) | 5 (83.3) | 3 (50.0) |
| > 50000 | 9 | 3 (33.3) | 3 (100.0) | 1 (33.3) |
| | | $p = 0.128$ | $p = 0.225$ | $p = 0.939$ |

*(Continued)*

**Table 1.** (Continued)

| Factors | Total | Alcohol drinking behavior (n = 9,566) | | |
|---|---|---|---|---|
| | | Current drinker, n (% of total) | Heavy episodic drinker, n (% of current drinkers) | Regular drinker, n (% of current drinkers) |
| Smoking status | | | | |
| No | 7167 | 1023 (14.3) | 706 (69.0) | 283 (27.7) |
| Yes | 2399 | 1615 (67.3) | 1244 (77.0) | 738 (45.7) |
| | | $p < 0.001^*$ | $p < 0.001^*$ | $p < 0.001^*$ |

[1] Chi-square test, IQR = interquartile range, THB = Thai Baht

$^*$ p < 0.05

### Relationship between alcohol sales licenses and proportion of alcohol consumption

The spatiotemporal mixed model analysis showed that an increase of one outlet per 1,000 population would increase the proportion of current drinkers among adolescents by 8% (odds ratio [OR] 1.08, 95% confidence interval [CI] 1.04–1.45), especially in male adolescents (OR 1.09, 95% CI 1.04–1.14) (Table 2). However, it affected HEDs only among female adolescents (OR 1.23, 95% CI 1.05–1.44).

In the survey year (temporal effect), the proportion of current drinkers in female adolescents continuously increased from 2007 to 2017 (for 2017, OR 2.49, 95% CI 1.66–3.75). The proportion of HEDs decreased by five times compared to 2007. In all the survey years, the drinking behaviors of current drinkers, HEDs, and regular drinkers were not associated with the SDI, except for female regular drinkers in the third quintile. They significantly consumed less alcohol than their counterparts in the first quintile, the richest participants (OR 0.19, 95% CI 0.06–0.65).

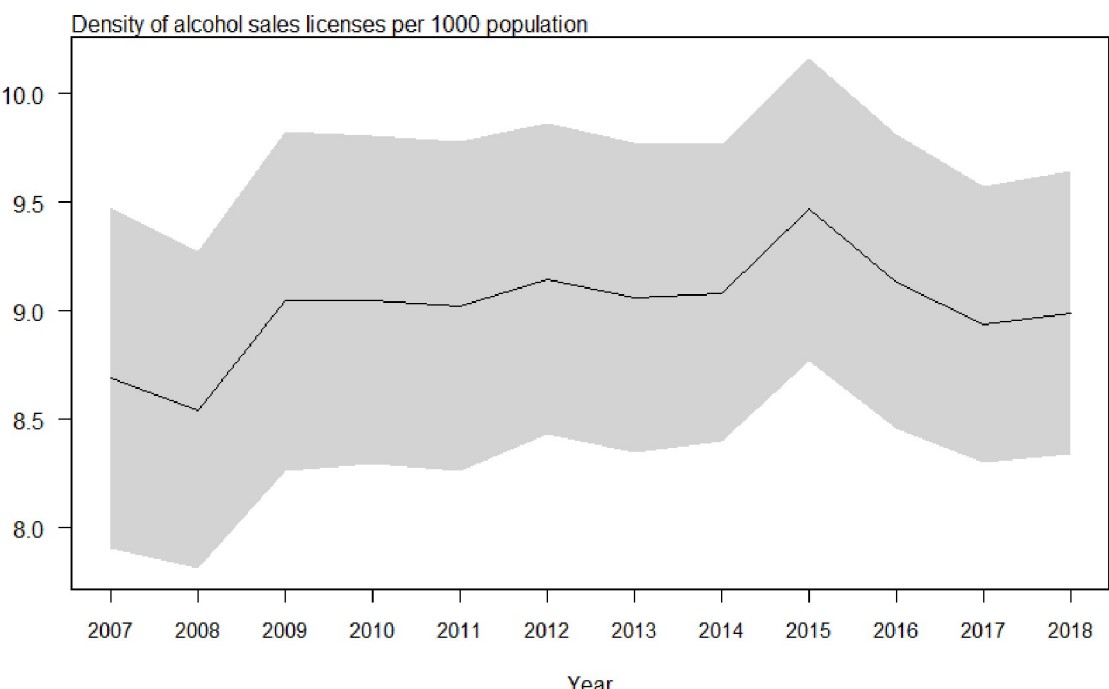

**Fig 1. Number of liquor licenses per 1,000 Inhabitants in 2007–2018 in Thailand.**

**Table 2. Association of density of alcohol sales licenses, survey year, and social deprivation index with proportion of alcohol consumption among Thai adolescents: Results from a spatiotemporal mixed model analysis (N = 9,566).**

| Factor | Drinking behavior, Adjusted OR (95% CI) | | | | | | | | |
|---|---|---|---|---|---|---|---|---|---|
| | Current drinker | | | Heavy episodic drinker | | | Regular drinker | | |
| | Male | Female | Total | Male | Female | Total | Male | Female | Total |
| Density of alcohol sales licenses per 1,000 population | | | | | | | | | |
| | *1.09** | 1.06 | *1.08** | 1.00 | *1.23** | 1.03 | 1.02 | 0.89 | 1.01 |
| | *(1.04–1.14)* | (0.97–1.16) | *(1.04–1.45)* | (0.96–1.05) | *(1.05–1.44)* | (0.99–1.07) | (0.97–1.06) | (0.78–1.02) | (0.96–1.05) |
| Survey year (ref: 2007) | | | | | | | | | |
| 2011 | 1.04 | *1.63** | 1.11 | 1.01 | 1.64 | 1.06 | 0.91 | 0.99 | 0.91 |
| | (0.90–1.2) | *(1.19–2.23)* | (0.98–1.25) | (0.8–1.27) | (0.86–3.15) | (0.86–1.32) | (0.76–1.1) | (0.49–2.01) | (0.76–1.09) |
| 2017 | 0.96 | *2.49** | *1.23** | *0.19** | *0.22** | *0.20** | 0.82 | 0.87 | 0.83 |
| | (0.79–1.15) | *(1.66–3.75)* | *(1.04–1.13)* | *(0.15–0.24)* | *(0.1–0.51)* | *(0.16–0.25)* | (0.64–1.05) | (0.35–2.13) | (0.66–1.05) |
| Social deprivation index (ref: Q1 or richest) | | | | | | | | | |
| Q2 | 0.90 | 1.31 | 0.97 | 0.98 | 1.91 | 1.11 | 1.23 | 1.17 | 1.24 |
| | (0.56–1.45) | (0.57–3.02) | (0.61–1.55) | (0.67–1.43) | (0.74–4.96) | (0.79–1.56) | (0.85–1.77) | (0.49–2.83) | (0.86–1.78) |
| Q3 | 1.19 | 1.07 | 1.27 | 0.89 | 0.87 | 0.95 | 1.17 | *0.19** | 1.09 |
| | (0.72–1.96) | (0.43–2.71) | (0.78–2.08) | (0.59–1.32) | (0.33–2.31) | (0.66–1.36) | (0.79–1.72) | *(0.06–0.65)* | (0.73–1.62) |
| Q4 | 1.18 | 1.27 | 1.26 | 0.78 | 2.20 | 0.89 | 0.90 | 0.81 | 0.89 |
| | (0.73–1.91) | (0.55–2.97) | (0.79–2.01) | (0.53–1.14) | (0.77–6.26) | (0.62–1.26) | (0.62–1.3) | (0.31–2.15) | (0.61–1.3) |
| Q5 (or poorest) | 0.96 | 0.84 | 1.06 | 0.97 | 0.57 | 0.99 | 0.88 | 0.40 | 0.85 |
| | (0.59–1.55) | (0.35–1.99) | (0.66–1.69) | (0.66–1.44) | (0.19–1.68) | (0.7–1.41) | (0.60–1.28) | (0.12–1.38) | (0.58–1.25) |

* This table presents the variables of interest from the multiple spatiotemporal mixed model analysis. Adjusted variables (i.e. region, household area, age group, educational level, marital status, monthly household income, and smoking status) included in the analysis but not shown here did not significantly contribute to the model.

OR = odds ratio

* $p < 0.05$

**Fig 2** presents the heatmaps for the number of alcohol sales licenses per 1,000 people and Thai adolescents' proportion of alcohol consumption, which are the percentages of current drinkers and HEDs in each province in 2007, 2011, and 2017. In the 10-year period, the density of alcohol sales licenses was high in the same provinces (e.g., Bangkok, Chiang Mai, Phuket, Surat Thani, and their surrounding areas). When compared visually on a year-by-year basis, the density of alcohol sales licenses and Thai adolescents' proportion of alcohol consumption showed no apparent relationship. Moreover, despite the overall trends in percentage and distribution of current drinkers being similar in all three waves, we found a decrease in heavy episodic drinking in 2017. The rate of heavy episodic drinking remained constant across all survey waves in provinces with previously high proportions.

## Discussion

### Principal findings and previous studies

We found that the density of alcohol sales licenses increased the likelihood of current adolescent drinking, especially in male participants. This result was consistent with our hypothesis that the higher the density, the more adolescents would drink. Density increased the odds of heavy episodic drinking in female adolescents. The average number of alcohol sales licenses from 2007 to 2018 was nine per 1,000 people. The density of alcohol sales licenses was higher in 2015 compared with 2007 and 2008. We concluded that the number of alcohol sales licenses

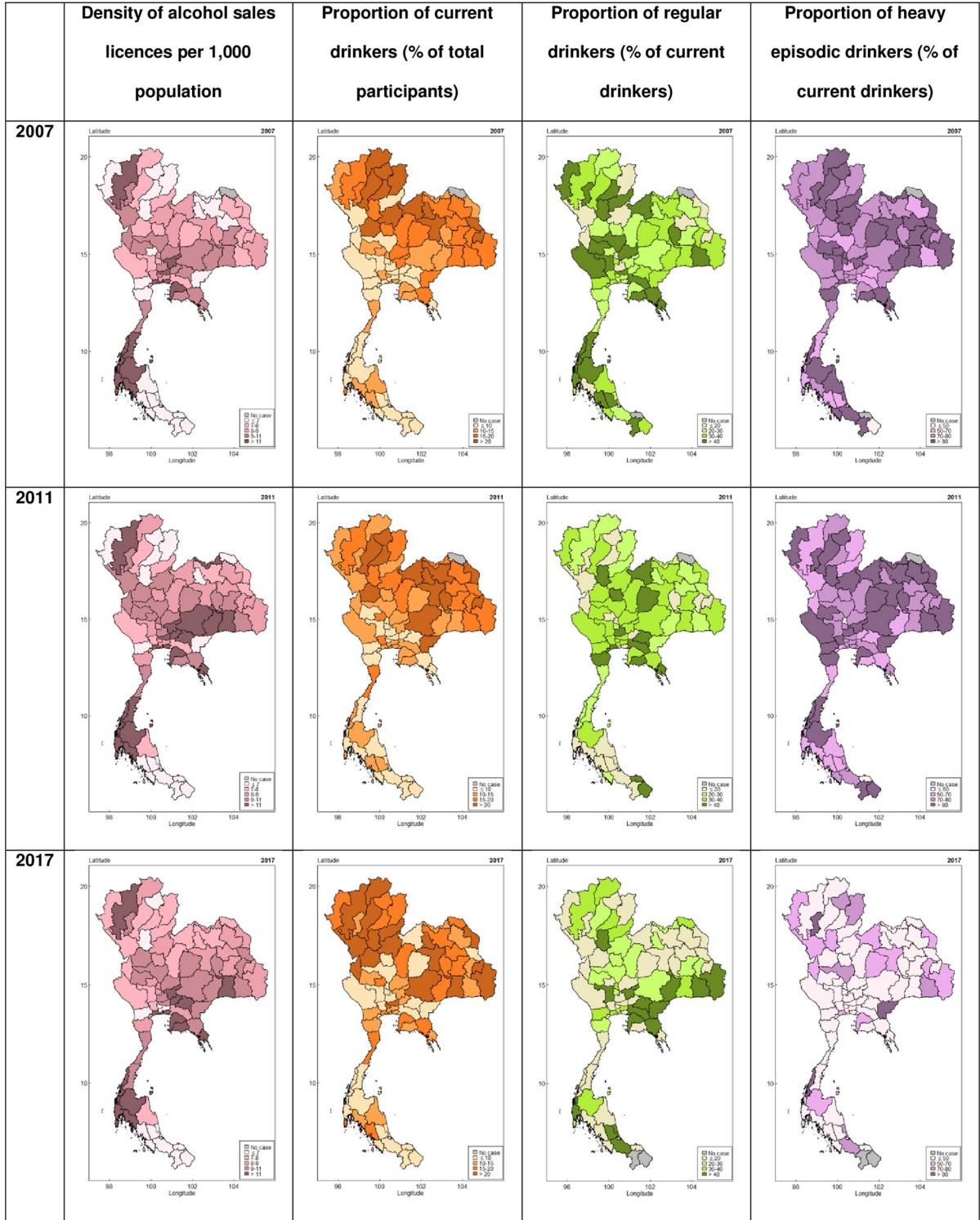

Note: Scales differ across the categories.

**Fig 2. Map of the density of alcohol sales licenses and proportion of alcohol consumption among Thai adolescents in 2007, 2011, and 2017.**

showed steady density adjustments and no decline. The number of alcohol sales licenses, at nine per 1,000 people, is alarmingly high compared to the average of 1.9 outlets per 1,000 adults in Japan [48] and 5.17 outlets per 1,000 residents in Wisconsin in the United States [49]. However, notably, the alcohol outlet licensing policy regarding the type of premises that need a license to sell alcohol differs from one jurisdiction to another.

The spatiotemporal mixed model analysis found no association between drinking patterns and the SDI in adolescents. However, existing research has suggested that adults with lower socioeconomic status tend to exhibit higher alcohol consumption compared to their higher socioeconomic counterparts [50, 51].

From the heatmap, we observed a markedly high density of alcohol outlets in Chon Buri Province in the central region, Chiang Mai Province in the northern region, Phuket Province, and Surat Thani Province in the southern region, as well as the surrounding areas of these four provinces in all three waves. These four provinces are well-known tourist destinations (e.g., Pattaya, located in Chon Buri Province) and ranked in the top 10 provinces with the most tourism revenue in 2017 [52]. However, we found fewer current drinkers in these areas compared to the other areas with lower alcohol outlet density. This result is in contrast to a previous study suggesting that the youth would be the population affected by the tourism drinking culture [53]. We hypothesized that this result is owing to industries concentrating more on tourists than on local residents, as evidenced by a previous study that found a high density of alcohol outlets in the vicinity of tourist attractions in Chon Buri Province [54].

Heavy episodic drinking among adolescents was associated with the density of sales licenses because the environment stimulates alcohol consumption. Paschall et al. proposed that this association was mediated by perceived alcohol availability and perceived approval of alcohol use [55]. Other studies have suggested that high outlet density was associated with alcohol supply by parents and the early initiation of alcohol consumption [56–58]. These mechanisms might also explain our results. Although some findings differ [59, 60], our results corroborate those of previous studies that reported an association between outlet density and alcohol consumption in New Zealand, Brazil, and Australia [26, 32, 61]. Moreover, considering the results from previous studies showing that Thai adolescents often purchase alcoholic beverages from local vendors [62, 63], it is clear that more outlets mean more access to alcohol, which likely leads to more consumption.

Although the density of alcohol sales licenses changed consistently over time, the total number did not decrease. This supports Thaikla's study [64], which found that the total number of alcohol outlets and alcohol outlet density between 2009 and 2011 were relatively stable. However, an increasing trend of alcohol outlets and the density of outlets were observed in 2014. This trend has also been reported in Canada and Los Angeles County in the United States [65, 66].

We found an increased incidence of heavy episodic drinking in female adolescents. Considering that the vigorous marketing efforts targeting women [67] and the modern Western lifestyle, characterized by shifted gender roles due to higher education and economic empowerment [68], influenced young Thai women to engage in heavy episodic drinking [69], we proposed that the abundant availability of alcohol could further encourage consumption. Nonetheless, our result contradicts data from Korea, where a decrease in alcohol consumption in the same group was observed [70]. In England, there has been a decline in alcohol consumption among female adolescents, albeit at a slower rate than in male adolescents [71]. A review of studies in high-income countries supports the findings of a declining trend in youth drinking that was more prevalent in male than female adolescents [72]. However, there is limited evidence regarding how female adolescent drinking has changed over the years in middle- and low-income countries with socioeconomic backgrounds similar to Thailand, despite the

pronounced increase in drinking in countries such as Vietnam [73]. It should be noted that the increased consumption in Thai female adolescents is concerning as an early onset of alcohol use correlates with a higher likelihood of heavy episodic drinking among Thai females. Those who begin drinking before the age of 20 show greater odds of engaging in heavy episodic drinking compared to those who start at 25 or older. This relationship appears stronger among females than males [74].

## Limitations and strengths

This study had several limitations. First, the causal relationship between the density of alcohol sales licenses and drinking behaviors may not provide a rational explanation. This is a common limitation in this type of research. Thus, the results of this study must be communicated cautiously to the public. Second, there were confounding factors because decisions on alcohol consumption might be influenced by factors other than access to sales licenses, such as types of outlets. As the types of outlets are not recorded in the registration form, our recommendation to the Excise Department is to collect these data. Further, the study included temporal relations and used information from a national survey of the National Statistical Office of Thailand, which dates back 10 years and does not concern prospective data. However, the researchers addressed the problem by collecting as many variables as possible—at both the individual and community levels—to mitigate these factors. Finally, we combined data from two sources, which can generate illusory relationships because of distinctive research methodologies and time lags. Thus, the information must be interpreted with caution.

This study's strength lies in identifying a provincial driving force. The research team presented provincial data on key variables over a decade, including the density of sales licenses in each province and the proportion of alcohol consumption by province. This provides a reference for related agencies in other provinces to proceed with a relevant mechanism. Another strength of this study is the relatively high reliability of the data owing to the information concerning sales licenses covering up to 89% of the total.

## Implications and further studies

Any policy and operational agency can apply our findings. The approach is relevant for the Office of the Alcoholic Beverage Control Committee, the Department of Disease Control, and the Ministry of Public Health. They can use our findings to make the Excise Department's alcohol licensing control laws stricter. In addition, the Stop Drinking Network can benefit from campaigning at the local level. This study reports on each province separately to emphasize spatial management or area-based intervention in promoting the management measures of its local people and government agencies. It also highlights the necessity for those who make legislative decisions to provide academic evidence that the density of alcohol sales outlets affects behavior. They should highlight the impact of alcohol consumption in Thailand and advocate for a rigorous review of the rules for issuing and renewing licenses. These measures could reduce the detrimental effects of having numerous distribution points. At a practical level, it is recommended that future research focus on a smaller geographic area, such as the provincial level, to examine the difference in each neighborhood. We believe that the smaller the area, the easier it is for local governments to tackle these problems.

We should exercise caution when referring to the causal relationship between the density of alcohol sales licenses and drinking behaviors using prospective data in future studies. Similarly, analyzing information by combining databases should be performed cautiously. As we could not identify the types of outlets (e.g., local vendors, bars, and restaurants) and their

relationship with consumption owing to our limited data, these topics are reserved for future work.

## Conclusion

We found a correlation between increased alcohol sales licenses and alcohol consumption among Thai adolescents. Therefore, government organizations should devise and implement targeted strategies to reduce the number of licenses issued for the sale of alcohol and limit adolescent alcohol consumption. Moreover, Thailand can reduce adolescent alcohol use by strengthening the SAFER strategies: enforcing stricter sales and age restrictions, enhancing drink-driving enforcement, expanding youth-targeted treatments, banning appealing alcohol ads to youths, and raising alcohol prices. These steps will make alcohol less accessible and appealing to young people.

## Acknowledgments

The authors wish to thank Editage and the International Affairs Department, Faculty of Medicine, Prince of Songkla University, for proofreading the manuscript.

## Author Contributions

**Conceptualization:** Polathep Vichitkunakorn.

**Data curation:** Polathep Vichitkunakorn, Sawitri Assanangkornchai, Kanittha Thaikla, Suhaimee Buya, Mfahmee Talib, Warangkhana Duangpaen, Warintorn Bunyanukul.

**Formal analysis:** Polathep Vichitkunakorn, Suhaimee Buya.

**Funding acquisition:** Polathep Vichitkunakorn, Sawitri Assanangkornchai, Kanittha Thaikla.

**Investigation:** Polathep Vichitkunakorn, Sawitri Assanangkornchai, Kanittha Thaikla.

**Methodology:** Polathep Vichitkunakorn, Sawitri Assanangkornchai, Kanittha Thaikla, Mfahmee Talib, Monsicha Sittisombut.

**Project administration:** Polathep Vichitkunakorn, Supeecha Rungruang.

**Resources:** Sawitri Assanangkornchai, Kanittha Thaikla.

**Supervision:** Sawitri Assanangkornchai.

**Validation:** Polathep Vichitkunakorn, Sawitri Assanangkornchai, Kanittha Thaikla, Suhaimee Buya.

**Visualization:** Suhaimee Buya.

**Writing – original draft:** Polathep Vichitkunakorn, Warangkhana Duangpaen, Warintorn Bunyanukul.

**Writing – review & editing:** Monsicha Sittisombut.

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
