## [Decision Letter · Decision Letter 0]

2 Apr 2024

PONE-D-23-42353Alcohol outlet density and adolescent drinking behaviors in Thailand, 2007–2017: A spatiotemporal mixed model analysisPLOS ONE

Dear Dr. Sittisombut,

Thank you for submitting your manuscript to PLOS ONE. After careful consideration, we feel that it has merit but does not fully meet PLOS ONE’s publication criteria as it currently stands. Therefore, we invite you to submit a revised version of the manuscript that addresses the points raised during the review process. Please submit your revised manuscript by May 17 2024 11:59PM. If you will need more time than this to complete your revisions, please reply to this message or contact the journal office at plosone@plos.org. Please include the following items when submitting your revised manuscript:A rebuttal letter that responds to each point raised by the academic editor and reviewer(s). You should upload this letter as a separate file labeled 'Response to Reviewers'.A marked-up copy of your manuscript that highlights changes made to the original version. You should upload this as a separate file labeled 'Revised Manuscript with Track Changes'.An unmarked version of your revised paper without tracked changes. You should upload this as a separate file labeled 'Manuscript'.

We look forward to receiving your revised manuscript.

Kind regards,

Larissa Loures Mendes, Ph.D.

Academic Editor

PLOS ONE

Journal Requirements:

2. In the online submission form, you indicated that The data underlying the results presented in this study are available upon request from the corresponding author.

4. We note that Figure 2 in your submission contain map images which may be copyrighted. All PLOS content is published under the Creative Commons Attribution License (CC BY 4.0), which means that the manuscript, images, and Supporting Information files will be freely available online, and any third party is permitted to access, download, copy, distribute, and use these materials in any way, even commercially, with proper attribution. For these reasons, we cannot publish previously copyrighted maps or satellite images created using proprietary data, such as Google software (Google Maps, Street View, and Earth). For more information, see our copyright guidelines: http://journals.plos.org/plosone/s/licenses-and-copyright.

We require you to either present written permission from the copyright holder to publish these figures specifically under the CC BY 4.0 license, or remove the figures from your submission:

Additional Editor Comments:

Dear Authors,

The reviewers have indicated a number of changes and suggestions in the manuscript and in the current format it cannot be considered for publication. I suggest revising the manuscript and submitting the revised version for a new analysis.

Sincerely, Larissa

Reviewers' comments:

Reviewer's Responses to Questions

**Comments to the Author**

1. Is the manuscript technically sound, and do the data support the conclusions?

Reviewer #1: Yes

Reviewer #2: Yes

2. Has the statistical analysis been performed appropriately and rigorously? 

Reviewer #1: Yes

Reviewer #2: Yes

3. Have the authors made all data underlying the findings in their manuscript fully available?

Reviewer #1: No

Reviewer #2: Yes

4. Is the manuscript presented in an intelligible fashion and written in standard English?

Reviewer #1: Yes

Reviewer #2: Yes

5. Review Comments to the Author

**Reviewer #1**: The study's aim was to explore the relationship between alcohol outlet density and drinking behaviors among Thai adolescents. The manuscript is well-written and has interesting methodology and results, especially to base actions to prevent and combat alcohol consumption by adolescents. However, some minor details need to be revised, mainly in the introduction, and the conclusion with the implications needs to be improved and present more advanced discussions on public policies.

Introduction

The introduction needs a revision of the information presented; readers from outside Thailand need to understand everything only by reading your manuscript. And I recommend including the hypothesis of your study after the objective.

Line 2-3: Are these efforts just in online settings?

Line 3-4: We do not know this Alcoholic Beverage Control Act B.E. 2551. What is it? A Thailand national law?

Line 18: We do not know this Excise Act B.E. 2560. What is it?

Line 20-23: Is this information important here? (about the types of permits)

Line 24: Which campus?

Line 27-37: I recommend reading again and adjusting the order in which the information appears. I could not understand much of your purpose with this paragraph; it is confusing.

Materials and Methods

I recommend you analyze if you can say your study has a temporal analysis.

Line 44: Your study is an analysis of secondary data, not a secondary analysis.

Line 45-46: Include information about the data of alcohol establishments.

Line 65: You need to make it clear that you collected information on establishments from 2007 to 2018, and that all the years have information. I only understood this in the results.

Reading the methodology, I had a question: was the density of establishments selling alcohol estimated for each province, and to associate it with consumption by teenagers, was this density analyzed by province? Or was it by the area around the houses?

Line 115-117: You must include all the variables analyzed. The readers do not go to table 1 to discover all the variables you include in the analysis.

Results

Figure 1: Include the location in the title of the figure.

Table 2: Include the total number of participants of your study in the title of the table.

Figure 2: Review the figure to see some mistakes, and I recommend writing a note explaining that the scales vary through the categories.

Discussion

Line 235-243: What is your hypothesis as to why these episodes of heavy drinking are more frequent among female adolescents?

Conclusion

Have you only thought about regulating licenses and limiting teenagers' consumption? I recommend a deeper and more advanced analysis of the issue. Wouldn't education actions with teenagers be necessary to raise awareness about alcohol consumption and its harms? Just limiting or prohibiting alcohol is not working, as this is already happening, and the results of your study show that teenagers still consume alcohol. I recommend reviewing the implications and suggestions (and changing the abstract as well).

**Reviewer #2:** At the beginning of the abstract the authors say: “This study aims to explore the relationship between alcohol outlet density and drinking behaviors among Thai adolescents” and at the end of the introduction they say: “This study examines the relationship between alcohol outlet density and the proportion of alcohol use among Thai teenagers.” Review the end of the introduction and standardize for consumer behavior. In the methodology (line 72) it mentions that they are drinking behaviors, and it is important to change the term “the proportion” at the end of the introduction

The study is confusing in determining a target audience, in the title and objective they mention that they are teenagers, but in the introduction (line 30) they mention young adults. It is important to define the age range of the group studied and standardize the terms

Line 106 -107: As potential confounding variables for each participant, some variables were included that I don't think apply to the lives of adolescents (study participants), such as marital status and monthly income. That's why it's important to make it clear who is being studied. If they are adults (which I don't believe they are) these adjustment variables are relevant. Other interesting adjustment variables for adolescents, for example: age, occurrence of bullying, difficulty interacting with peers, satisfaction with school life, alcohol consumption by someone in the family

Line 115: The confounding variables described can be found in lines 106, 107 and 108.

In the introduction, an approach to the impacts of alcohol consumption on the health of adolescents/young adults would be interesting.

6. PLOS authors have the option to publish the peer review history of their article (what does this mean?). If published, this will include your full peer review and any attached files.

Reviewer #1: **Yes: **Luana Lara Rocha

Reviewer #2: No

---

## [Author Response · Author response to Decision Letter 0]

17 May 2024

PONE-D-23-42353

Response to Reviewers

Dear Dr. Mendes, 

Thank you for giving us the opportunity to submit a revised draft of the manuscript “Alcohol outlet density and adolescent drinking behaviors in Thailand, 2007–2017: A spatiotemporal mixed model analysis” for publication in PLOS ONE. We appreciate the time and effort that you and the reviewers dedicated to providing feedback on our manuscript and are grateful for the insightful comments on and valuable improvements to our paper. We have incorporated most of the suggestions provided by the reviewers. These changes are marked-up within the manuscript using tracked changes. Please see the blue part below for a point-by-point response to the reviewers’ comments and concerns. All page numbers refer to the revised manuscript file with tracked changes. 

Journal Requirements: 

Author response: Thank you for your comment. We have checked that our manuscript meets the target journal’s requirements.

2. In the online submission form, you indicated that The data underlying the results presented in this study are available upon request from the corresponding author. 

Author response: Due to ethical limitations governing the dissemination of a de-identified dataset, it is not permissible to share the data publicly. We assert that the manuscript in its present form includes all necessary information pertinent to our study. Researchers who fulfill the criteria for accessing confidential data are urged to contact the corresponding author to facilitate discussions regarding their requests for data access.

3. Please include your full ethics statement in the ‘Methods’ section of your manuscript file. In your statement, please include the full name of the IRB or ethics committee who approved or waived your study, as well as whether or not you 

obtained informed written or verbal consent. If consent was waived for your study, please include this information in your statement as well. 

Author response: Thank you for your comment. We have incorporated the ethics statement in the Methods section accordingly. 

The revised text reads as follows in the “Study design” section from lines 122-125: 

“The authors have applied for an amendment regarding protocols for analyses of anonymized secondary data. All obtained data were anonymous. Ethical approval was obtained from the Human Research Ethics Committee of the Faculty of Medicine, Prince of Songkla University (REC. 62-054-18-1).”

4. We note that Figure 2 in your submission contain map images which may be copyrighted. All PLOS content is published under the Creative Commons Attribution License (CC BY 4.0), which means that the manuscript, images, and Supporting Information files will be freely available online, and any third party is permitted to access, download, copy, distribute, and use these materials in any way, even commercially, with proper attribution. 

For these reasons, we cannot publish previously copyrighted maps or satellite images created using proprietary data, such as Google software (Google Maps, Street View, and Earth). For more information, see our copyright guidelines: http://journals.plos.org/plosone/s/licenses-and-copyright. 

We require you to either present written permission from the copyright holder to publish these figures specifically under the CC BY 4.0 license, or remove the figures from your submission: 

In the figure caption of the copyrighted figure, please include the following text: “Reprinted from (Drefs) under a CC BY license, with permission from [name of publisher], original copyright [original copyright year].” 

USGS EROS (Earth Resources Observatory and Science (EROS) Center) (public domain): http://eros.usgs.gov/# Natural Earth (public domain): http://www.naturalearthdata.com/

Author response: Thank you for pointing this out. The map in Figure 2 was created using QGIS, Open Source Geographic Information System (GIS) licensed under the GNU General Public License. Hence, we believe that we hold the copyright to the maps. Additionally, we have revised a sentence in the Statistical analysis section to give a clearer explanation of this topic.

The revised text reads as follows in the “Statistical analysis” section from lines 210-211: 

“We created the map using “QGIS,” an open source Geographic Information System (GIS).”

If you have any further questions or require clarification on this matter, please do not hesitate to let us know. 

5. Please include captions for your Supporting Information files at the end of your manuscript, and update any in-text citations to match accordingly. Please see our Supporting Information guidelines for more information: 

http://journals.plos.org/plosone/s/supporting-information. 

Additional Editor Comments: 

Author response: There is no supporting information file due to the reason mentioned above.

Dear Authors,

The reviewers have indicated a number of changes and suggestions in the manuscript and in the current format it cannot be considered for publication. I suggest revising the manuscript and submitting the revised version for a new analysis.

Sincerely, Larissa 

Reviewers' comments:

Reviewer's Responses to Questions

Comments to the Author 

1. Is the manuscript technically sound, and do the data support the conclusions? 

Reviewer #1: Yes Reviewer #2: Yes 

2. Has the statistical analysis been performed appropriately and rigorously? 

Reviewer #1: Yes Reviewer #2: Yes 

3. Have the authors made all data underlying the findings in their manuscript fully available? 

Reviewer #1: No Reviewer #2: Yes 

4. Is the manuscript presented in an intelligible fashion and written in standard English? 

Reviewer #1: Yes Reviewer #2: Yes 

5. Review Comments to the Author 

Reviewer #1: The study's aim was to explore the relationship between alcohol outlet density and drinking behaviors among Thai adolescents. The manuscript is well-written and has interesting methodology and results, especially to base actions to prevent and combat alcohol consumption by adolescents. However, some minor details need to be revised, mainly in the introduction, and the conclusion with the implications needs to be improved and present more advanced discussions on public policies. 

Author response: Thank you for your valuable suggestion. We agree with your comment and have incorporated your suggestions throughout the manuscript. 

Introduction 

The introduction needs a revision of the information presented; readers from outside Thailand need to understand everything only by reading your manuscript. 

Author response: Thank you for your valuable suggestion. Accordingly, throughout the introduction, we have included additional information regarding Thailand’s alcohol control laws and the current situation of alcohol consumption in Thailand, as well as modified some parts of the introduction to facilitate better logic.

The revised text reads as follows in the “Introduction” section from lines 27-28: 

“Alcohol is a major threat to the individual and society as a whole. The effects on individual health range from cardiovascular diseases and infectious diseases to injuries. It can lead to suicide and interpersonal violence.” 

lines 53-66:

“The World Health Organization has introduced SAFER strategies as alcohol control interventions to reduce harmful alcohol use. The strategies consist of the following: (1) Strengthen restriction on alcohol availability; (2) Advance and enforce drunk driving countermeasures; (3) Facilitate access to screening, brief interventions, and treatment; (4) Enforce bans or comprehensive restrictions on alcohol advertising, sponsorship, and promotion; and (5) Raise prices on alcohol through excise taxes and pricing policies.

Currently, Thailand has three major national laws that are in accordance with the SAFER strategies: the Alcoholic Beverage Control Act B.E. 2551 (2008), Excise Act B.E. 2560 (2017), and Road Traffic Act B.E. 2522 (1979) [11]. The Alcoholic Beverage Control Act B.E. 2551 (2008) is Thailand's first legislative effort aimed specifically at controlling the consumption and distribution of alcoholic beverages. The Excise Act B.E. 2560 (2017) regulates the taxation and licensing of production, and the sales and import of products and services, including alcoholic products. In the context of alcohol regulation, the Road Traffic Act B.E.2522 (1979) sets limits on the alcohol blood content (BAC) for drivers.”

lines 71-72:

“Moreover, a study has shown that Thai adolescents are inclined to start drinking earlier than in the past.”

And I recommend including the hypothesis of your study after the objective. 

Author response: Thank you for this suggestion. We agree with your comment. Therefore, we have included the hypothesis as follows on lines 104-105:

“Our hypothesis was as follows: the higher the alcohol density, the higher the proportion of adolescents’ alcohol use.”

We have also incorporated the interpretation of our results based on the hypothesis in the “Discussion” section on lines 271-272.

“This result was consistent with our hypothesis that the higher the density, the more adolescents would drink.”

Line 2-3: Are these efforts just in online settings? 

Author response: Thank you for pointing this out. We have revised this sentence to provide more insight into the marketing efforts of the alcohol industry and their effects.

The revised text reads as follows on lines 50-52: 

“To exacerbate the situation, there has been an increase in various marketing efforts from the alcohol industry, including mass media commercials, in-store displays, merchandise, and online marketing.”

Line 3-4: We do not know this Alcoholic Beverage Control Act B.E. 2551. What is it? A Thailand national law? 

Author response: Thank you for pointing this out. We have included additional information about the Alcoholic Beverage Act B.E. 2551. 

The revised text reads as follows on lines 59-63: 

“Currently, Thailand has three major national laws that are in accordance with the SAFER strategies: the Alcoholic Beverage Control Act B.E. 2551 (2008), Excise Act B.E. 2560 (2017), and Road Traffic Act B.E. 2522 (1979). The Alcoholic Beverage Control Act B.E. 2551 (2008) is Thailand's first legislative effort aimed specifically at controlling the consumption and distribution of alcoholic beverages. …”

Line 18: We do not know this Excise Act B.E. 2560. What is it? 

Author response: Thank you for pointing this out. We have included additional information about the Excise Act B.E. 2560.

The revised text reads as follows on lines 59-64: 

“…Currently, Thailand has three major national laws that are in accordance with the SAFER strategies: the Alcoholic Beverage Control Act B.E. 2551 (2008), Excise Act B.E. 2560 (2017) and Road Traffic Act B.E. 2522 (1979). …. The Excise Act B.E. 2560 (2017) regulates the taxation and licensing of production, and the sales and import of products and services, including alcoholic products. …”

Line 20-23: Is this information important here? (about the types of permits) 

Author response: Thank you for pointing this out. We originally included this information to provide background on the type of sale permits, which would be mentioned later in the strengths and limitations section. However, upon review, we have determined that this information is not necessary here. Thus, we have removed it.

Line 24: Which campus? 

Author response: Thank you for pointing this out. Details regarding “campus” from the literature have been added on line 87. 

“A st

---

## [Decision Letter · Decision Letter 1]

18 Jul 2024

Alcohol outlet density and adolescent drinking behaviors in Thailand, 2007–2017: A spatiotemporal mixed model analysis

PONE-D-23-42353R1

Dear Dr. Sittisombut,

We’re pleased to inform you that your manuscript has been judged scientifically suitable for publication and will be formally accepted for publication once it meets all outstanding technical requirements.

Kind regards,

Larissa Loures Mendes, Ph.D.

Academic Editor

PLOS ONE

Additional Editor Comments (optional):

Reviewers' comments:

Reviewer's Responses to Questions

**Comments to the Author**

1. If the authors have adequately addressed your comments raised in a previous round of review and you feel that this manuscript is now acceptable for publication, you may indicate that here to bypass the “Comments to the Author” section, enter your conflict of interest statement in the “Confidential to Editor” section, and submit your "Accept" recommendation.

Reviewer #1: All comments have been addressed

Reviewer #2: All comments have been addressed

2. Is the manuscript technically sound, and do the data support the conclusions?

Reviewer #1: Yes

Reviewer #2: Yes

3. Has the statistical analysis been performed appropriately and rigorously? 

Reviewer #1: Yes

Reviewer #2: Yes

4. Have the authors made all data underlying the findings in their manuscript fully available?

Reviewer #1: Yes

Reviewer #2: No

5. Is the manuscript presented in an intelligible fashion and written in standard English?

Reviewer #1: Yes

Reviewer #2: Yes

6. Review Comments to the Author

Reviewer #1: The manuscript review "Alcohol outlet density and adolescent drinking behaviors in Thailand, 2007–2017: A spatiotemporal mixed model analysis" was carried out in full, and all suggestions and questions were addressed and resolved. I consider that the manuscript can be published.

Reviewer #2: (No Response)

7. PLOS authors have the option to publish the peer review history of their article (what does this mean?). If published, this will include your full peer review and any attached files.

Reviewer #1: **Yes: **Luana Lara Rocha

Reviewer #2: No

---

## [Editor Report · Acceptance letter]

12 Aug 2024

PONE-D-23-42353R1 

PLOS ONE

Dear Dr. Sittisombut, 

I'm pleased to inform you that your manuscript has been deemed suitable for publication in PLOS ONE. Congratulations! Your manuscript is now being handed over to our production team.

Kind regards, 

on behalf of

Dr. Larissa Loures Mendes 

Academic Editor

PLOS ONE